

# The stochastic skeleton model for the Madden-Julian Oscillation with time-dependent observation-based forcing

Noémie Ehstand[1,2], Reik V. Donner[3,4], Cristóbal López[1], Marcelo Barreiro[5], and Emilio Hernández-García[1]

[1]Instituto de Física Interdisciplinar y Sistemas Complejos (IFISC), CSIC-UIB. Campus Universitat de les Illes Balears, E-07122 Palma de Mallorca, Spain
[2]Present address: Institut Terre et Environnement de Strasbourg (ITES), Université de Strasbourg, 5 rue Descartes, Strasbourg F-67084, France
[3]Department of Water, Environment, Construction and Safety, Magdeburg-Stendal University of Applied Sciences, Breitscheidstraße 2, D-39114 Magdeburg, Germany
[4]Research Department IV-Complexity Science, Potsdam Institute for Climate Impact Research (PIK) - Member of the Leibniz Association, Telegrafenberg A31, D-14473 Potsdam, Germany
[5]Departamento de Ciencias de la Atmósfera y Física de los Oceanos, Instituto de Física, Facultad de Ciencias, Universidad de la República, Igua 4225, 11400 Montevideo, Uruguay

**Correspondence:** Emilio Hernández-García (emilio@ifisc.uib-csic.es)

**Abstract.** We analyze solutions to the stochastic skeleton model, a minimal nonlinear oscillator model for the Madden-Julian Oscillation (MJO). This model has been recognized for its ability to reproduce several large-scale features of the MJO. In previous studies, the model's forcings were predominantly chosen to be mathematically simple and time-independent. Here, we present solutions to the model with time-dependent observation-based forcing functions. Our results show that the model, with these more realistic forcing functions, successfully replicates key characteristics of MJO events, such as their lifetime, extent, and amplitude, whose statistics agree well with observations. However, we find that the seasonality of MJO events and the spatial variations in the MJO properties are not well reproduced. Having implemented the model in the presence of time-dependent forcings, we can analyze the impact of temporal variability at different time scales. In particular, we study the model's ability to reflect changes in MJO characteristics under the different phases of ENSO. We find that it does not capture differences in studied characteristics of MJO events in response to differences in conditions during El Niño, La Niña, and neutral ENSO.

## 1 Introduction

The Madden–Julian Oscillation (Madden & Julian, 1972) is the dominant component of intra-seasonal variability in the tropics (Woolnough, 2019). It is a planetary-scale wave envelope of smaller-scale convective processes, slowly propagating eastward along the equator, with a period of 40 to 50 days on average, and a mean propagation speed of about 5 m/s, with fluctuations ranging from 1 to 9 m/s (Chen & Wang, 2020). As it propagates, the MJO causes disturbances in rainfall and winds which impact weather and climate in both the tropics and the extratropics (Zhang, 2013). In particular, the MJO plays a significant role in the occurrence of various weather extremes such as tropical cyclones, tornadoes, extreme rainfall events and extreme



surface temperatures (Jones et al., 2004; Pohl & Camberlin, 2006; Juliá et al., 2012; Thompson & Roundy, 2013; Zhang, 2013;
Jeong et al., 2005).

With its slow eastward propagation on intraseasonal time scales, the MJO is a key source of subseasonal predictability (Woolnough, 2019; Vitart et al., 2019) and is of utter importance to enhance preparedness for such extreme events. Accurately forecasting the MJO has thus become a foremost goal in subseasonal-to-seasonal forecasting. But, even with significant progress in recent years (Lim et al., 2018; Kim et al., 2019; Silini et al., 2021), its representation and predictability in numeri-
cal models continues to be a challenging task (Liu et al., 2024). Moreover, a recent study suggests that there is an increased probability of extinction of the MJO after 27 days, which seems to point to an internal mechanism of exhaustion rather than to the effect of an external barrier (Corral et al., 2023).

The underlying physical mechanisms that govern the MJO are not yet fully understood (Zhang, 2013). A deeper understanding of these mechanisms is however essential for improving MJO predictions, as they form the foundation for the development
of more accurate models. This knowledge gap has motivated the development of data-driven models (Díaz et al., 2023) and simplified low-dimensional physics-based models (Zhang et al., 2020) that aim to capture the key aspects of the MJO's behavior. One such model, known as the skeleton model, provides a framework for investigating some essential dynamics of the MJO.

The skeleton model is a minimal nonlinear oscillator model for the Madden-Julian Oscillation, introduced by Majda &
Stechmann (2009b). The model is derived from the well-known primitive equations for the dry atmosphere in the tropics complemented with a modelling of moist convection. It is an idealized model with a minimal number of parameters and a single quadratic nonlinearity. Nonetheless, it provides insights into atmospheric dynamics when coupled with moist convection in the tropics and is able to reproduce large-scale features of the MJO, including its slow eastward propagation at about 5m/s, its near-zero group velocity and its horizontal quadrupole vortex structure. Several versions of the model have been developed based
on the initial work from Majda and Stechmann (Majda & Stechmann, 2011; Thual et al., 2014; Thual & Majda, 2015, 2016). In particular, adding stochasticity to the initial model, Thual et al. (2014) showed that the skeleton theory also captures the intermittent generation of MJO events and their organization into wave trains with growth and decay.

The model includes two forcing functions, representing latent heating and radiative cooling in the tropics. In the majority of previous studies, these functions were chosen to be mathematically simple, time-independent and identical (Majda & Stech-
mann, 2009b, 2011; Thual et al., 2014; Thual & Majda, 2016; Stachnik et al., 2015; Chen & Stechmann, 2016). The aim of the present study is to assess whether the model forced with more dynamic and observationally grounded functions can still accurately reproduce key characteristics of MJO events. By using more realistic forcings, our work offers a more robust test of the skeleton model's applicability and relevance in understanding the MJO dynamics as, simplified, static forcings might limit the realism of the model's outputs.



Precisely, we consider the stochastic skeleton model as presented in Thual et al. (2014) with forcing functions computed from observational and reanalysis data following a methodology presented in Ogrosky & Stechmann (2015). Further, while Ogrosky & Stechmann (2015) forced the model with long-term averages of the computed functions, in this work, we keep their time-dependence. Thus, we present solutions to the model when the latent heating and radiative cooling functions are observation-based, time-dependent, non-identical functions. To our knowledge this is the first study of the MJO skeleton

model with forcing functions combining these three characteristics. We show that several properties of the simulated events (in particular, their lifetime, extent and amplitude) agree well with the ones of observed MJO events. On the other hand, we find that the seasonality of MJO events and the spatial variations in the MJO properties are not well reproduced by the model.

Further, several studies suggested that the sea surface temperature (SST) variability at interannual and longer time scales, in particular that associated with the El Niño–Southern Oscillation (ENSO), modulates certain characteristics of the MJO such

as the extent of its eastward propagation (Kessler, 2001; Tam & Lau, 2005; Pohl & Matthews, 2007), its lifetime (Pohl & Matthews, 2007) and its speed (Wei & Ren, 2019; Díaz et al., 2023). With our implementation of the skeleton model using time-dependent forcings, we can now explore the ability of the MJO skeleton model to induce differences in selected MJO characteristics under El Niño, La Niña and neutral ENSO conditions.

This paper is organized as follows. In Section 2 we present the model and in Section 3 the computation of the forcing

functions from observational and reanalysis data is explained. The identification of MJO events in the model is performed based on an objective index: the skeleton multivariate MJO (SMM) index. The computation of the SMM index is described in Section 4. The numerical solutions to the model are then presented in Section 5. The identification of MJO events is briefly illustrated in Section 6. Then, in Section 7, we compare statistics of MJO characteristics between observations and simulations. Finally, in Section 8, we present the statistics of selected characteristics of MJO events in observations and in the stochastic

skeleton model simulations under El Niño, La Niña and neutral ENSO conditions.

## 2    The MJO skeleton model

### 2.1    Deterministic model

The MJO skeleton model (Majda & Stechmann, 2009b, 2011) combines the linear, long-wave scaled, primitive equations (see White, 2003; Vallis, 2017) with a conservation equation for the moisture and a dynamic equation describing the interactions





between the lower tropospheric moisture anomaly and the planetary-scale envelope of convective activity:

$$u_t - yv = -p_x,$$
$$yu = -p_y,$$
$$0 = -p_z + \theta,$$
$$u_x + v_y + w_z = 0,$$
$$\theta_t + w = \bar{H}a - s^\theta,$$
$$q_t - \tilde{Q}w = -\bar{H}a + s^q,$$
$$a_t = \Gamma qa, \tag{1}$$

where $x$, $y$, and $z$ are the zonal, meridional and vertical coordinates, and $u$, $v$ and $w$ are the velocity anomalies in these directions, respectively, $p$ and $\theta$ are the pressure and potential temperature anomalies, $q$ is the lower tropospheric moisture anomaly, and $a$ is the envelope of convective activity. Note that all variables are anomalies from a radiative-convective equi-

librium except for $a$. The variables $s^\theta$ and $s^q$ represent external sinks/sources of temperature and moisture, such as radiative cooling and latent heating, respectively. They act as forcing in the model and will be described in more detail in Section 3. The model has only three parameters: $\tilde{Q}$ is the mean background vertical moisture gradient, $\Gamma$ represents the sensitivity of convective activity tendency to moisture anomalies and $\bar{H}$ is a scaling constant for the convective activity. The equations have been non-dimensionalized using some standard equatorial length and time scales (Majda & Stechmann, 2009a). The first five

equations of the model describe the dry dynamics of the atmosphere (the conservation of horizontal momentum, the hydrostatic balance, the conservation of mass and the conservation of potential temperature). The sixth equation describes the conservation of low level moisture. The last equation is the non-linear interaction between moisture and convection. It entails the idea that the moisture anomalies influence the growth and decay rates of the planetary-scale envelope of convective activity.

To obtain the model in its simplest version (Majda & Stechmann, 2009b, 2011; Thual et al., 2014), Equations (1) are trun-

cated in the vertical and meridional directions. In the vertical, the variables are expanded in terms of sines and cosines, keeping only the first baroclinic mode, i.e. $u(x,y,z,t) \approx u_1(x,y,t)\sqrt{2}\cos(z)$, $v(x,y,z,t) \approx v_1(x,y,t)\sqrt{2}\cos(z)$, $p(x,y,z,t) \approx p_1(x,y,t)\sqrt{2}\cos(z)$, $w(x,y,z,t) \approx w_1(x,y,t)\sqrt{2}\sin(z)$, $\theta(x,y,z,t) \approx \theta_1(x,y,t)\sqrt{2}\sin(z)$ (see Khouider et al., 2013). Here, $z \in [0,\pi]$ in non-dimensional units, and $z \in [0, H_{top}]$ in dimensional units, where $H_{top}$ is the height of the tropopause. Further, it is assumed that $q = q_1(x,y,t)\sqrt{2}\sin(z)$, $a = a_1(x,y,t)\sqrt{2}\sin(z)$, $s^\theta = s_1^\theta(x,y,t)\sqrt{2}\sin(z)$ and $s^q = s_1^q(x,y,t)\sqrt{2}\sin(z)$

(see Majda & Tong, 2016). Dropping the subscript 1 for simplicity (i.e. $u_1 \to u$, $v_1 \to v$, etc.), System (1) becomes:

$$u_t - yv - \theta_x = 0,$$
$$yu - \theta_y = 0,$$
$$\theta_t - u_x - v_y = \bar{H}a - s^\theta,$$
$$q_t + \tilde{Q}(u_x + v_y) = -\bar{H}a + s^q,$$
$$a_t = \Gamma qa. \tag{2}$$





In the meridional direction, the variables and forcing functions are expanded using parabolic cylinder functions $\{\phi_m(y)\}$ such that, $u(x,y,t) = \sum_m u_m(x,t)\phi_m(y)$, etc., where the first three modes have the form $\phi_0(y) = \pi^{-1/4}\exp(-y^2/2)$, $\phi_1(y) = \pi^{-1/4}\sqrt{2}y\exp(-y^2/2)$, $\phi_2(y) = \pi^{-1/4}(1/\sqrt{2})(2y^2-1)\exp(-y^2/2)$. This expansion facilitates a change of variable in the dry dynamics (rows 1-4 of Eq. (2)) which allows to introduce new variables representing equatorial waves. In the simplest version of the model, only the amplitudes of the first mode of equatorial Kelvin wave structure ($K$) and equatorial Rossby wave structure ($R$) are kept, defined as $K \equiv \frac{u_0 - \theta_0}{\sqrt{2}}$ and $R \equiv u_2 - \theta_2 - \frac{u_0 + \theta_0}{\sqrt{2}}$. In addition, it is assumed that the envelope of convection/ wave activity $a$ takes the form $a(x,y,t) = A(x,t)\phi_0(y) = [A_s(x) + A^*(x,t)]\phi_0(y)$ with $A_s$ representing the background state and $A^*$ fluctuations around this state, and that $q(x,y,t)$, $s^\theta(x,y,t)$ and $s^q(x,y,t)$ are truncated at the first mode: $q(x,y,t) = Q(x,t)\phi_0(y)$, $s^\theta(x,y,t) = S^\theta(x,t)\phi_0(y)$ and $s^q(x,y,t) = S^q(x,t)\phi_0(y)$. The final truncated equations then read:

$$K_t + K_x = -\frac{1}{\sqrt{2}}(\bar{H}A - S^\theta),$$

$$R_t - \frac{1}{3}R_x = -\frac{2\sqrt{2}}{3}(\bar{H}A - S^\theta),$$

$$Q_t + \frac{1}{\sqrt{2}}\tilde{Q}K_x - \frac{1}{6\sqrt{2}}\tilde{Q}R_x = \frac{\tilde{Q}}{6}(\bar{H}A - S^\theta) - (\bar{H}A - S^q),$$

$$A_t = \gamma\Gamma Q(A_s + A^*),\tag{3}$$

where all the functions depend now only on $x$ and $t$, and $\gamma = \int(\phi_0)^3 dy \approx 0.6$ results from the meridional projection of the non-linear equation.

The variables $u$, $v$ and $\theta$ can be approximately recovered via

$$u(x,y,t) = \frac{1}{\sqrt{2}}\left[K(x,t) - \frac{1}{2}R(x,t)\right]\phi_0(y) + \frac{1}{4}R(x,t)\phi_2(y),$$

$$v(x,y,t) = \left[\frac{1}{3}\partial_x R(x,t) - \frac{1}{3\sqrt{2}}(\bar{H}A(x,t) - S^\theta(x,t))\right]\phi_1(y),$$

$$\theta(x,y,t) = -\frac{1}{\sqrt{2}}\left[K(x,t) + \frac{1}{2}R(x,t)\right]\phi_0(y) - \frac{1}{4}R(x,t)\phi_2(y).\tag{4}$$

## 2.2 Stochastic model

In the skeleton model, the MJO is initiated and sustained by the synoptic (sub-planetary) scale convective activity patterns, which are considered collectively via their planetary-scale envelope $a$. These synoptic scale processes include for instance deep convective clouds which are highly irregular, intermittent and with low predictability. To account for such processes, Thual et al. (2014) proposed a modified version of the skeleton model, where the last equation in (1) is replaced by a stochastic process. The authors showed that this *stochastic skeleton model* is able to generate intermittent MJO wave trains with growth and decay as observed in reality. Specifically, the variable $a$ in System (1) is replaced by a random variable taking discrete values separated by $\Delta a$, that is $a = \eta\Delta a$, with $\eta \in \mathbb{N}$. The evolution of $a$ is controlled by a birth-death process allowing for intermittent transitions between states $\eta$. This process is described by the following master equation for the probability of $\eta$, $P(\eta, t)$:

$$\partial_t P(\eta) = [\lambda(\eta - 1)P(\eta - 1) - \lambda(\eta)P(\eta)] + [\mu(\eta + 1)P(\eta + 1) - \mu(\eta)P(\eta)]\tag{5}$$





where $\lambda$ is the upward rate of transition and $\mu$ the downward rate. The choice of $\lambda$ and $\mu$ is made such that the dynamics of the non-stochastic skeleton model is recovered on average (see Thual et al., 2014).

## 3 Observation-based time-dependent forcing functions

As mentioned in the introduction, the majority of previous studies on the MJO skeleton model used idealized, time-independent and equal forcing functions $s^\theta$ (radiative cooling) and $s^q$ (latent heating), i.e. $s^\theta(\mathbf{x}) = s^q(\mathbf{x})$ (see Majda & Stechmann, 2009b, 2011; Thual et al., 2014; Thual & Majda, 2016; Stachnik et al., 2015; Chen & Stechmann, 2016). Thual et al. (2015) first studied the solutions of the skeleton model with periodic variations in the forcing. The authors used an idealized warm pool state representation of $s^\theta(\mathbf{x},t) = s^q(\mathbf{x},t)$ migrating seasonally in the meridional direction. In Ogrosky & Stechmann (2015) 130 and later in Ogrosky et al. (2017), the authors computed the forcing functions based on long-term means of observational and reanalysis data, leading to more realistic functions where $s^\theta(\mathbf{x}) \neq s^q(\mathbf{x})$.

Here, we consider the stochastic skeleton model, with forcing functions computed from observational and reanalysis data following the methodology presented in Ogrosky & Stechmann (2015). However, unlike Ogrosky and Stechmann, we do not take long-term averages but consider monthly varying data. We are hence concerned with solutions of model (1) when 135 $s^\theta = s^\theta(\mathbf{x},t)$, $s^q = s^q(\mathbf{x},t)$ and $s^\theta \neq s^q$, that is, when the forcings are realistic (observation-based), time-dependent and non-identical. This section explains the computation of the profiles.

### 3.1 Data sources

To estimate the forcing terms, we use NCEP/NCAR reanalysis latent heat net flux (Kalnay et al., 1996) for the computation of the latent heating $s^q$ and NCEP Global Precipitation Climatology Project (GPCP) data (Adler et al., 2017; Huffman et al., 140 2001) for the computation of $\bar{H}a$ (which then enters in the calculation of both $s^q$ and $s^\theta$ as explained below). The chosen data sub-sets cover the period 1979-2021 with a monthly resolution. Both fields have global spatial coverage, with a resolution of $1.875° \times 1.875°$ (degrees of latitude and longitude) for the latent heat flux data set and $2.5° \times 2.5°$ for the precipitation data set.

### 3.2 Estimation procedure

First, following Ogrosky & Stechmann (2015), the 2D field $\bar{H}a(x,y)$ is computed using the formula

$$\bar{H}a = \left( \frac{g\rho_w L_v}{p_0 c_p} \right) M, \tag{6}$$

where $M$ [m] represents the monthly precipitation data, $g = 9.8$ m/s$^2$ is the gravitational acceleration constant, $\rho_w = 10^3$ kg/m$^3$ is the density of water, $L_v = 2.5 \cdot 10^6$ J/kg is the latent heat of vaporization, $c_p = 1006$ J/(kg·K) is the specific heat of dry air at constant pressure, and $p_0 = 1.013 \cdot 10^5$ kg m$^{-1}$ s$^{-2}$ is the mean atmospheric pressure at mean sea level. The above formula 150 describes the rate at which the temperature of a column of air increases from the energy released by precipitation at a given location. To obtain the 1D equatorial profile needed for the truncated model (see Section 2, Eq. (3)), the 2D field is projected



onto the leading meridional mode $\phi_0$:

$$\bar{H}A(x,t) = \int_{-\infty}^{\infty} \bar{H}a(x,y,t)\phi_0(y)dy. \tag{7}$$

Second, still following Ogrosky & Stechmann (2015), we compute the 1D equatorial forcing profile $S^q(x,t)$. The latent heat
flux (LHF) is projected onto $\phi_0$:

$$LHF_0(x,t) = \int_{-\infty}^{\infty} LHF(x,y,t)\phi_0(y)dy, \tag{8}$$

and $S^q$ is computed according to Ogrosky & Stechmann (2015) as

$$S^q = H_{LHF} \cdot LHF_0, \tag{9}$$

where

$$H_{LHF} \approx \frac{\langle \bar{H}A \rangle_{t,x}}{\langle LHF_0 \rangle_{t,x}} \approx 0.0067 \text{ K day}^{-1}(\text{Wm}^{-2})^{-1},$$

with $\langle \cdot \rangle_{t,x}$ representing the time and zonal mean, implying that $\langle \bar{H}A \rangle$ balances $\langle S^q \rangle$.

In fact, according to Ogrosky & Stechmann (2015) (Eq. (7) in that paper), for a steady-state solution to exist in the skeleton
model, which has no damping, we must have

$$\langle \bar{H}A_s \rangle_x = \langle S^q_s \rangle_x = \langle S^\theta_s \rangle_x, \tag{10}$$

where $\langle \cdot \rangle_x$ represents the zonal mean and the subscript $s$ indicates the background state of the quantities, defined as their
long-time average. From Eq. (9) we see that this is satisfied for $\bar{H}A$ and $S^q$.

In addition, as explained in Ogrosky & Stechmann (2015) (Eq. (8) in that paper), the model background convective activity
must satisfy

$$\bar{H}A_s = \frac{S^q_s - \tilde{Q}S^\theta_s}{1 - \tilde{Q}}. \tag{11}$$

To make sure that this is the case, we compute $S^\theta(x,t)$ as:

$$S^\theta = \frac{1}{\tilde{Q}}S^q - \frac{(1-\tilde{Q})}{\tilde{Q}}\bar{H}A. \tag{12}$$

Note that, computed in this way, $S^\theta$ also automatically satisfies condition (10).

The computed functions $A_s$, $S^q$ and $S^\theta$ can be decomposed into spatial Fourier modes. In order to focus on planetary-scale
variations, only the first 8 Fourier modes are kept.

One of the aims of this work is to study the solutions to the MJO skeleton model when the forcing functions are *time-
dependent* observation-based functions. Therefore, while Ogrosky & Stechmann (2015) used long-term averages of the com-
puted $S^q$ and $S^\theta$, here we skip this step and keep the time dependence of the profiles. Nonetheless, we make sure to smooth



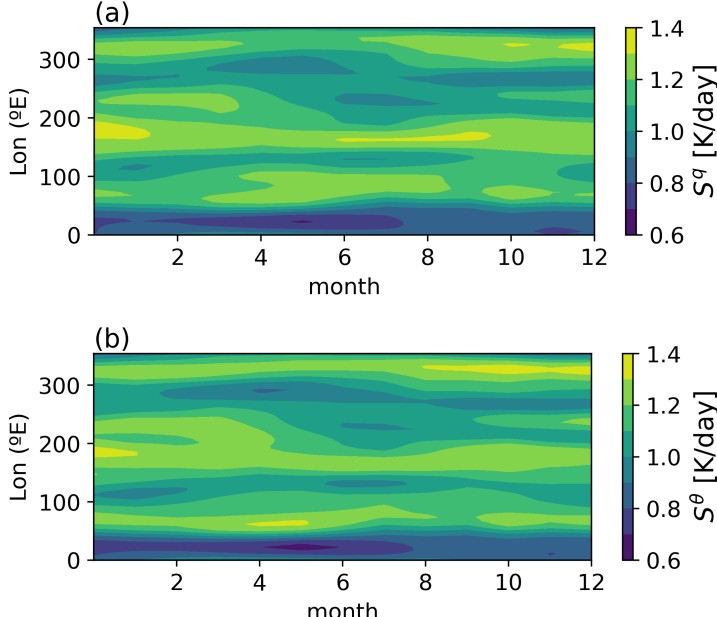

**Figure 1.** Evolution of the latent heating profile $S^q$ estimated from latent heat flux and the radiative cooling profile $S^\theta$ computed according to Eq. (12). The abscissa indicates months of the year 1979.

the functions in time in order to avoid sharp changes which could disrupt the model's statistical equilibrium. Specifically, as mentioned above, the data sets have monthly resolution. In order to smooth the temporal variation of the forcing, we compute a 3-month running mean of the forcing functions. The forcing profiles are then interpolated to the time step of the model. We hence obtain smooth time-dependent observation-based forcing functions $S^q$ and $S^\theta$. As an example, their evolution over the year 1979 is illustrated in Fig. 1.

## 4 Identification of MJO events: the skeleton multivariate MJO index

In observations, the most commonly used index to monitor the MJO is the real-time multivariate MJO (RMM) index developed by Wheeler & Hendon (2004). It is based on an empirical orthogonal function (EOF) analysis of the daily observed Outgoing Longwave Radiation (OLR) as well as lower tropospheric (850 hPa) and upper tropospheric (200 hPa) zonal winds, averaged meridionally in the equatorial region between 15°S-15°N. The index is typically represented in a two-dimensional phase space defined by the two dominant principal components (RMM1 and RMM2). This space is divided into 8 sectors (quadrants), each corresponding to a distinct phase of the MJO cycle as its convective center travels from the Indian Ocean across the Pacific and towards the Western Hemisphere.



In order to objectively identify MJO structures in the model output, we compute an index similar to the RMM index following
the methodology presented in Stachnik et al. (2015). The main difference is that while RMM uses three variables (upper and
lower tropospheric winds and OLR), the model index is based on a bivariate analysis with the (model) zonal wind $u$ as a direct
substitute for the lower tropospheric wind and the negative convective heating $-\bar{H}a$ as a proxy for OLR. Since clouds affect the
radiation emitted at the top of the atmosphere, OLR is often chosen as an indicator of cloudiness. Wheeler & Hendon (2004)
indeed used OLR as a proxy for convective activity, justifying the choice of $-\bar{H}a$ herein.[1] The steps for the computation of
this *skeleton multivariate MJO* (SMM) index are briefly described below and full details can be found in Stachnik et al. (2015).

We first isolate the intraseasonal signal in the data. To do so the daily anomalies of $u$ and $-\bar{H}a$ are filtered using a $20-100$
days Lanczos filter. Second, each field is normalized by its global standard deviation so that they have an equal contribution in
the computation of empirical orthogonal functions. Lastly, as in Wheeler & Hendon (2004), the model principal components
SMM1 and SMM2 are computed by projecting the filtered data onto the two leading EOF modes and standardizing the output
such that a value of unity represents an anomaly of 1 standard deviation from the mean.

Strong MJO activity is characterized by an index amplitude greater than or equal to 1. Precisely, following Stachnik et al.
(2015), MJO events or episodes are defined from the (SMM1,SMM2) time series as periods during which the following
conditions are met:

- The amplitude of SMM is greater than 1: $\sqrt{SMM1^2 + SMM2^2} \geq 1$.

- The propagation of the event is almost continually counterclockwise in the (SMM1, SMM2) space, corresponding to
  an almost continually eastward propagation (a westward propagation is limited to at most a single phase of the MJO
  cycle).

- The event propagates through at least 4 phases of the MJO cycle.

## 5 Numerical solution

In this section, we present the main features of the numerical solution to the stochastic MJO skeleton model (3) with (dimen-
sionless) parameters $\tilde{Q} = 0.9$, $\Gamma = 1.0$, $\bar{H} = 0.22$, and observation-based time-dependent sources of cooling and moistening
$S^\theta$ and $S^q$. Our Julia implementation of the model is available from Ehstand (2025). To make sure that the solutions are
presented for a statistically equilibrated regime, we run simulations for 215 years with forcing corresponding to the 43-year
period 1979-2021 repeated 5 times ($5 \times 43 = 215$). We then keep only the last 43 years which we consider representative of
the 1979-2021 period.

---

[1]Stechmann & Majda (2015) showed that OLR variations are proportional to the total diabatic cooling variations in the atmosphere. This might be
approximated in the model as the negative of the sum of latent (convective) heating and radiative cooling $-(\bar{H}a - s^\theta)$. Nonetheless, we choose to use $-\bar{H}a$
alone as a proxy for OLR, since the essential aim is to represent the equatorial convective activity.





### 5.1 Hovmöller diagrams of the model variables

The evolution of the skeleton model equatorial profiles for the lower tropospheric wind $u(x,t)$ and envelope of convective activity $\bar{H}a(x,t)$, computed from Eq. (4) with $y = 0$, are shown in Figure 2. The time axis represents one year of simulation with forcing profiles representative of the year 2005. Figure 2(a-b) represents the raw output data. Figure 2(c-d) shows the data

after filtering in time and space as to isolate planetary-scale intraseasonal variations. Precisely, the daily anomalies from the long-term mean have been filtered in time using a $20 - 100$ days Lanczos filter and smoothed in space by retaining only modes with Fourier wave number $k \leq 4$. While the non-filtered plots [Fig. 2(a,b)] highlight the small scale propagating waves, the filtering [Fig. 2(c,d)] allows to capture larger-scale features.

Westward propagating modes are well visible in panel (a), as well as in panel (c) and (d), e.g. around days 260 to 300. With

periods ranging from around 25 to 90 days, these modes could be related to equatorial Rossby waves (see also Section 5.2). On the other hand, large-scale eastward propagating waves are visible in panel (c) and (d), especially towards the beginning of the period, from day 0 to 150. These intraseasonal large-scale waves are likely associated with the MJO as we will see in Section 5.2. Two waves, one propagating eastward and one propagating westward, have been marked in Figure 2(d) with their respective phase speed. Finally, we observe that in the convective activity plots [Fig. 2(b,d)], a higher activity can be seen in

the region from $60°$E to $200°$E which corresponds to a region between the Indian Ocean and the western Pacific, which is the region where the MJO signal is usually the strongest.

### 5.2 Power spectrum of the envelope of convective activity

Figure 3 shows the zonal wavenumber - frequency power spectrum of the simulated (unfiltered) envelope of convective activity $\bar{H}a$. The zonal wavenumbers are expressed as multiples of $2\pi/40000$ km (40000 km being approximately the circumference of

the Earth) and frequencies are in cycles per day (cpd). The dashed lines indicate the 90 and 30 days periods. The MJO appears as an horizontally elongated high power structure in the zonal wavenumber spectrum with $1 \leq k \leq 5$, that is as a planetary-scale wave, with intraseasonal frequencies $1/90 \leq \omega \leq 1/30$ cpd and eastward propagation ($\omega/k > 0$). This structure has a dispersion relation $d\omega/dk \approx 0$ which is a typical characteristic of the MJO and is known to be reproduced well by the skeleton model (Majda & Stechmann, 2009b, 2011; Thual et al., 2014). The mean phase speed of the waves associated with this structure

(calculated as the mean of $\omega/k$ for the points with $\omega \in [1/90, 1/30]$ cpd, $k \in [1,5]$ and log-power greater than $-4.0$) is $\approx 5$ ms$^{-1}$. This MJO signal is visible in Figure 2(d) between days 1 and 150. In addition to the MJO signal there is also a high power structure at intraseasonal time scales with westward propagation. Previous studies have shown that these modes share some, although incomplete, features with convectively coupled equatorial Rossby waves and they have been referred to as moist Rossby modes (Majda & Stechmann, 2011; Thual et al., 2014). An example of such westward wave is visible in the filtered

convective activity in Figure 2(d) between days 260 and 300. At higher frequencies, $\omega > 0.06$ corresponding to periods shorter than 16 days, the high power peaks might be associated with dry Kelvin and dry Rossby modes (Majda & Stechmann, 2011; Thual et al., 2014). Overall the spectrum of $\bar{H}a$ agrees well with previous studies of the skeleton model (Thual et al., 2014;



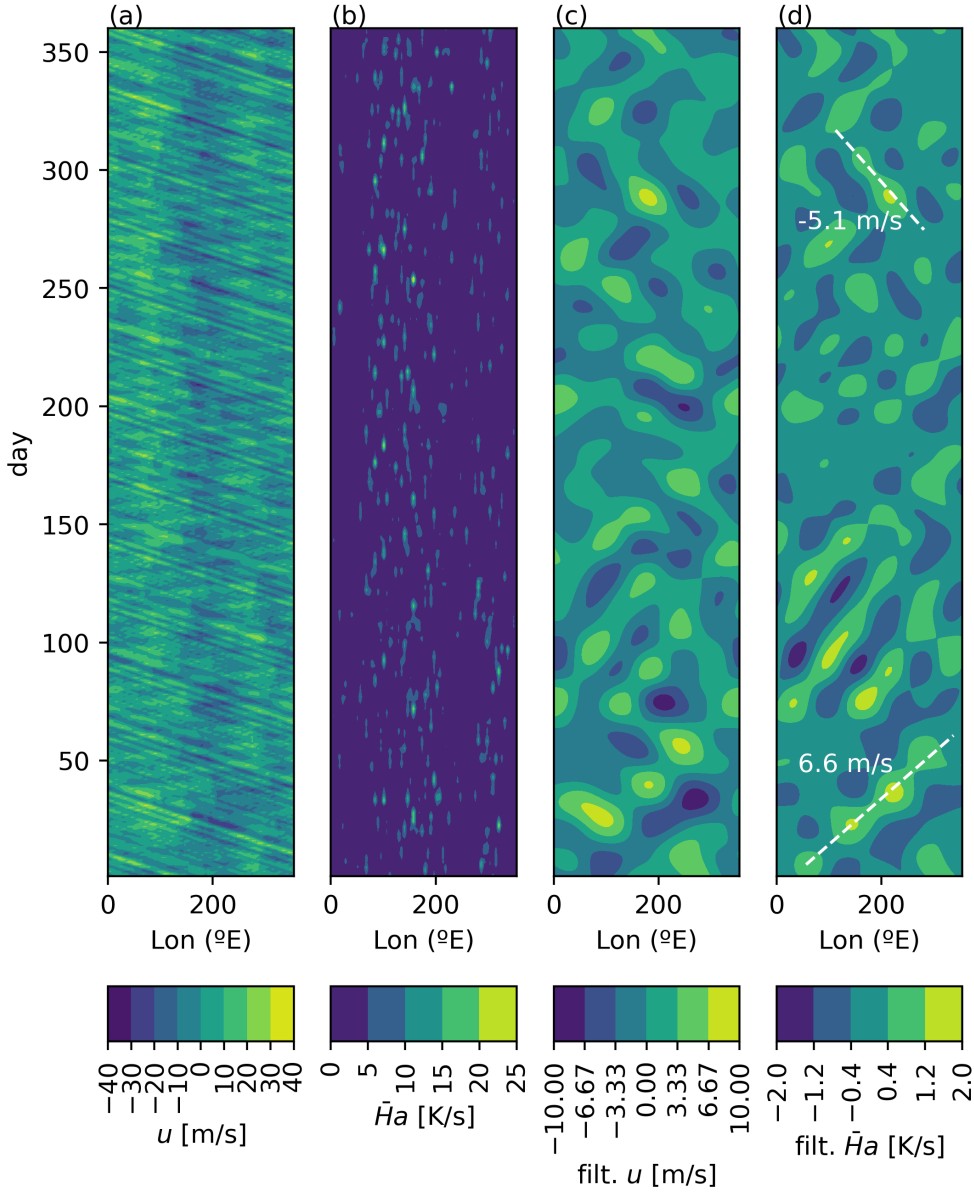

**Figure 2.** Hovmöller diagrams of the skeleton model lower tropospheric wind $u(x, y = 0, t)$ and envelope of convective activity $\bar{H}a(x, y = 0, t)$ at the equator. *(a,b)* raw data, *(c,d)* daily anomalies from the long-term mean, filtered in time and space as described in the text. One westward moving signal (likely of a moist Rossby wave) and one eastward moving signal (likely MJO activity) are marked in white in panel *(d)*.

Ogrosky & Stechmann, 2015). While several equatorial modes and especially the MJO are well represented, many modes are not reproduced by the model due to its minimal design, for instance convectively coupled Kelvin waves (Kiladis et al., 2009).





## 5.3 Climatology and variance of the envelope of convective activity

The long-term means of $\bar{H}a$ estimated from daily observations and simulated in the skeleton model are shown in Figure 4(a). Qualitatively, they agree well. However, the variance of $\bar{H}a$ is overestimated in the model as can be seen in Figure 4(b). Nonetheless, if we consider only the first 14 spatial modes, the variance is well reproduced by the model. This might be explained by the fact that the equations of the skeleton model are long-wave scaled, as the model is only concerned with planetary scales, and hence it might not be well suited to represent modes with higher wavenumbers.

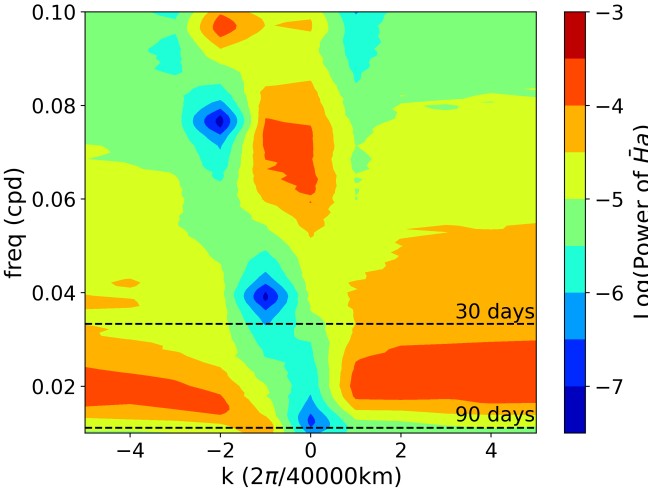

**Figure 3.** Zonal wavenumber - frequency power spectrum of the simulated envelope of convective activity $\bar{H}a$ (in base 10 - logarithm). The dashed lines mark the 90 and 30 days periods.

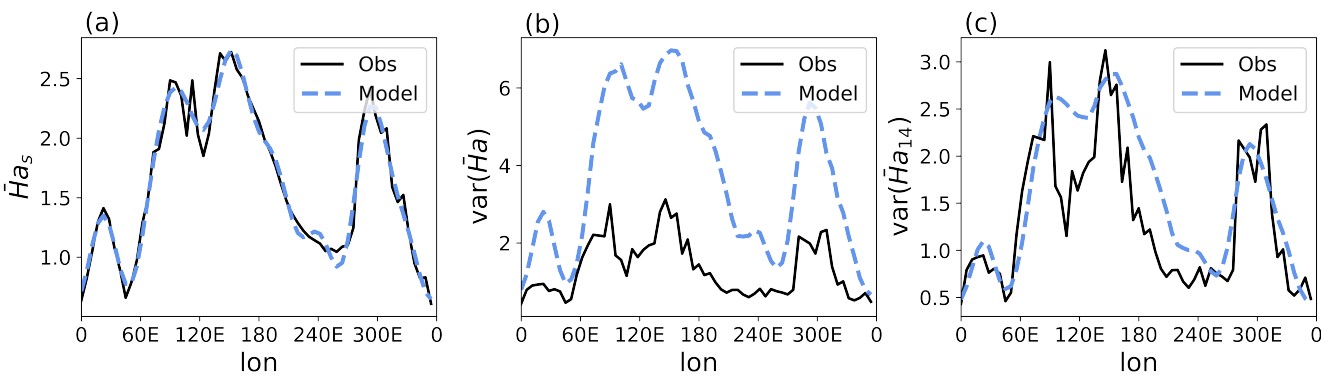

**Figure 4.** *(a)* Long-term averages of observed and modeled $\bar{H}a$ (envelope of convective activity). *(b)* Variance of modeled and observed $\bar{H}a$ when all spatial modes from the model output are kept. *(c)* Variance of modeled and observed $\bar{H}a$ when only the first 14 spatial modes from the model output are kept. Note the different scales in panels *(b)* and *(c)*.





# 6    Identification of MJO events with the SMM index

The results presented above suggest the presence of intermittent MJO wave structures propagating eastward. In order to objectively identify these structures, we use the skeleton multivariate MJO (SMM) index presented in Section 4.

An example of model SMM values over a 52-day period are shown in (SMM1, SMM2) phase space in Figure 5. The first and
last point of the series are annotated. The dark blue points satisfy the MJO event's criteria given in Section 4. Overall the MJO propagation is relatively smooth. As explained in Stachnik et al. (2015), this is partly due to the filtering of high frequencies in the model SMM index computation. This filtering eliminates some of the day to day variability and noise that are not removed in the computation of observation-based RMM index from Wheeler & Hendon (2004).

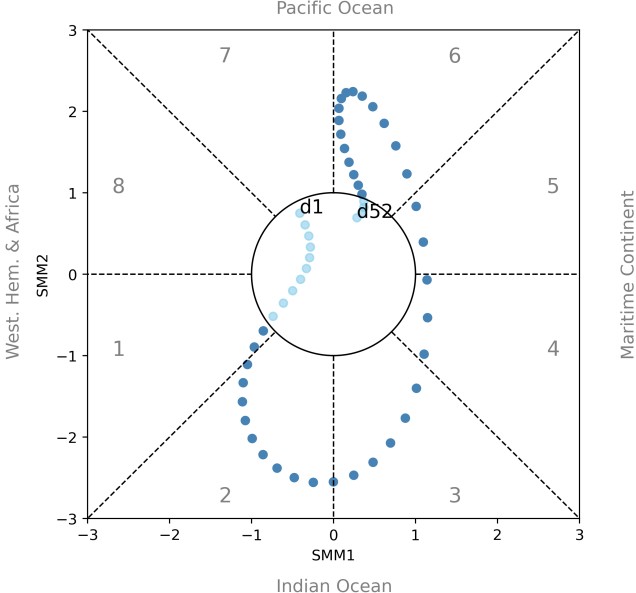

**Figure 5.** Phase-space diagram of the model SMM values for a 52-day period from a simulation forced with observation-based functions. The dots correspond to daily values of (SMM1, SMM2). The first day of the series is labelled 'd1', the last is labeled 'd52'. The circle in the center of the plot has unit radius, indicating the threshold at which the amplitude of SMM exceeds 1. Points in dark blue indicate an MJO event as defined by the criteria in Section 4.

# 7    MJO characteristics in the skeleton model and in observations

In the following, we study different characteristics of the MJO events in simulations and observations. The aim is to assess the ability of the MJO skeleton model to statistically reproduce the characteristics of observed MJO events. Here we list the chosen characteristics.





- – The *seasonal variation* in the occurrence of MJO events is obtained by recording the number of MJO events occurring (that is starting, continuing or ending) during each month of the year, where events are defined according to the criteria listed in Section 4.

- – The *duration* of an event is defined as the number of days from the first to the last day of the event.

- – We measure the *total angle* covered by an event tracked in the (SMM1, SMM2) phase space, that is the angle covered between the first and the last day of the event. This can roughly be assimilated to the "distance" covered by that event as it propagates along the equator.

- – The *maximum value of SMM amplitude*, where the amplitude is defined as $\sqrt{SMM1^2 + SMM2^2}$, is recorded for each event.

- – Finally, the *starting and ending phase of each event* (1-8) are recorded.

The model MJO events are identified using the SMM index and the criteria described in Section 4. We perform 15 independent simulation runs (in statistically equilibrated regime) with forcing profiles representative of the period 1979-2021, leading to a total of 980 modeled events. The observed events are computed according to the same criteria using the RMM index values which are freely available on the website of the Australian Bureau of Meteorology (http://www.bom.gov.au/climate/mjo/). We find 153 observed events over the period 1979-2021. For these observed events, the computation of the characteristics listed above is made from the (RMM1,RMM2) values.

## 7.1 MJO seasonal variations

We first look at the seasonal variation of MJO occurrences in the model and in observations in Figure 6. Observations show that MJO events are more frequent during boreal winter and spring, from December through May. In the simulations however no variation is detected. Recall that the forcing profiles have been averaged with a 3-month window. As a result the seasonal variations in heating and moistening are smoother and the differences between different seasons might be lost.

## 7.2 MJO lifetime, extent in SMM/RMM phase space and maximum SMM/RMM amplitude

We now compare the duration of MJO events, total angle covered in SMM/RMM phase space and maximum SMM/RMM amplitude in the model outputs and in observations. The cumulative probability distributions of these three characteristics are shown in Figure 7. As above, the statistics are based on 153 observed events over the period 1979-2021 and 980 modeled events from 15 independent simulations, run with forcing profiles representative of the same period. The duration of events in Figure 7(a), total angle (distances) covered by simulated events in Figure 7(b) and the maximum amplitude of SMM/RMM in Figure 7(c) compare well with observations. The average MJO event lifetime is $39.6 \pm 0.8$ days for the simulation and $36.1$ days for observations. This small difference is likely explained by the absence of certain sources of MJO termination in the minimalistic skeleton model leading to slightly longer events (although the longest event overall, of 153 days, occurs in observational data).




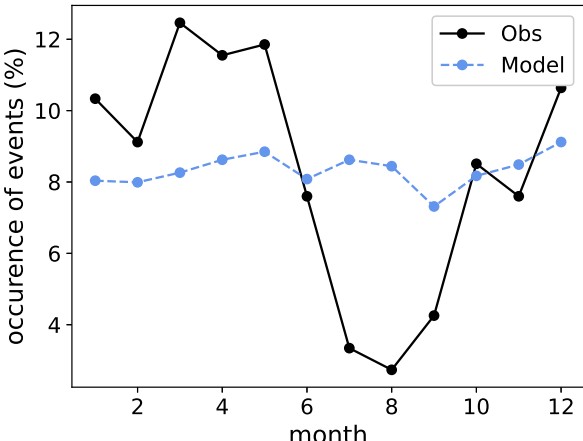

**Figure 6.** Occurrence of MJO events as a function of the month of the year

The mean angle is $(0.75 \pm 0.01) \cdot 2\pi$ for simulations and $0.75 \cdot 2\pi$ for observations. The mean of the maxima of SMM amplitude is $2.53 \pm 0.02$ for the simulations and the mean of the maxima of RMM amplitude $2.50$ for observations.

For each of the characteristics, a two-sample Kolmogorov-Smirnov test was conducted to compare the cumulative distribution functions obtained from observations and from the model. For the duration, despite apparent similarities, the obtained $p$ value is $0.0006$, suggesting that the two samples might come from different distributions. For the total angle covered in the SMM/RMM phase space, the test yields a $p$ value of $0.0428$. Thus, the test does not provide sufficient evidence to reject the null hypothesis that the model and observations produce the same distribution at significance level $0.04$. For the maxima of

SMM/RMM amplitude values, the $p$ value is $0.8116$, indicating again that the model and observations produce statistically indistinguishable distributions.

### 7.3    MJO starting and ending phases

Figure 8 shows the distribution of initial and final phases of MJO events. The error bar for a given bin is calculated using a binomial proportion confidence interval dependent on the pass and fail rate of recorded locations being assigned to that

particular bin. The dashed line indicates the equal likelihood of the ending location of an event being recorded in any of the 8 bins. Almost all error bars overlap this line, both in the simulations and observations, indicating that these graphs do not allow for statistically significant conclusions to be drawn. We note nonetheless that the distribution of starting phases might have some similarities. Two local maxima are observed around phases 2/3 and 6/7 for the starting location (although a high peak is also shown in phase 5 for the model which is not present in observations). For the ending phases, in observations, most of the

events end in phase 8, whereas in the model the peak in phase 8 is relatively small.

To conclude Section 7, the model cannot reproduce well seasonal variations in the MJO occurrences. Neither does it seem to be able to reproduce spatial differences in the MJO properties such as its starting and ending phase, although more investigation



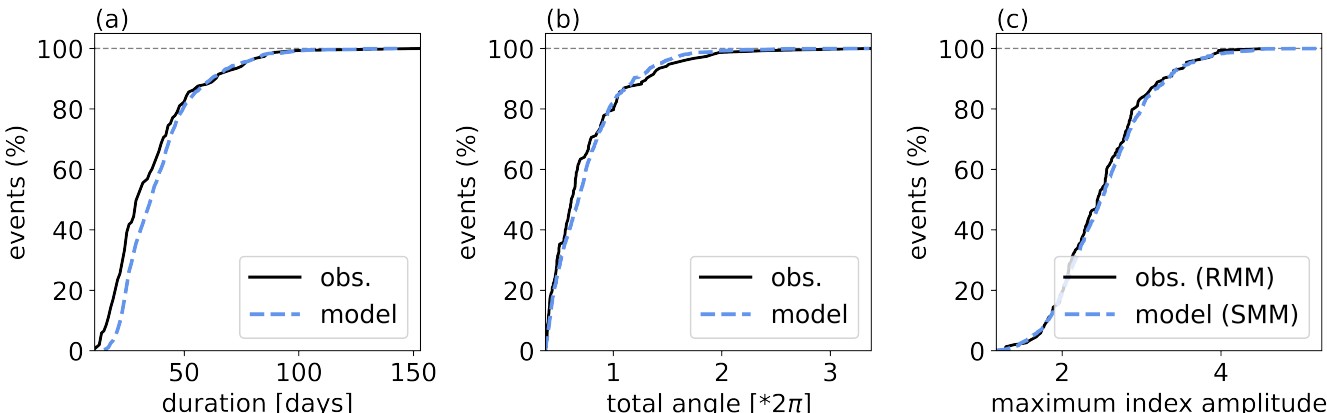

**Figure 7.** Cumulative probability distribution of the MJO events' durations, total angle covered in (RMM1, RMM2) / (SMM1, SMM2) phase space and maximum RMM/SMM value for observed and simulated MJO events (981 simulated events and 153 observed events).

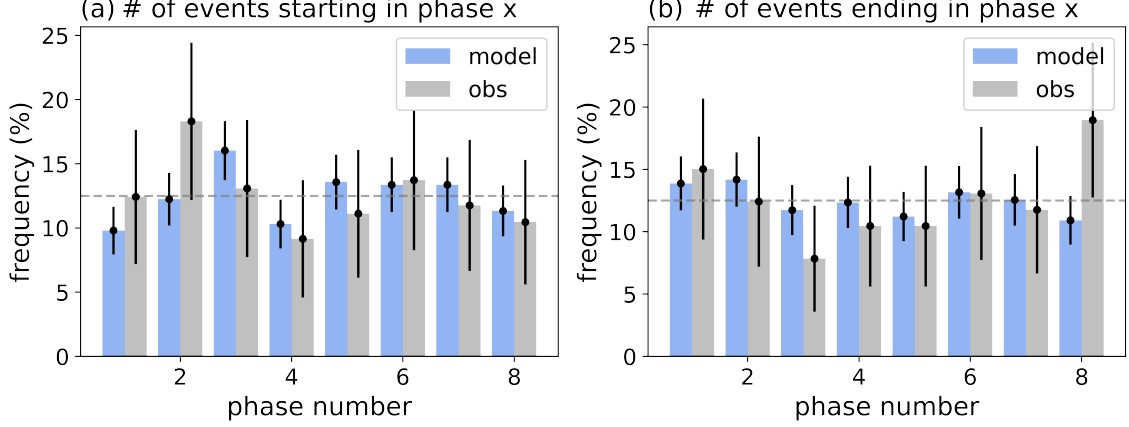

**Figure 8.** Histograms of the recorded starting *(a)* and ending phases *(b)* of MJO events in the model and in observations. The dashed line indicates the equal likelihood of the ending location of an event being recorded in any of the 8 bins. The error bars indicate the 98% uncertainty estimate, calculated from a binomial proportion confidence interval.





would be needed to establish the presence of statistically significant differences. Despite certain disparities in the cumulative distributions of events' characteristics, we consider that the model represents fairly well the duration, total angle covered in
SMM/RMM phase space and the maxima of SMM/RMM values. In the next section, we assess the ability of the model to produce differences in these last three characteristics under El Niño, La Niña and neutral ENSO conditions. Note that despite the model not being able to reproduce seasonal variations, MJO events might still be influenced by ENSO variations since they occur on much longer time scales (typically of 2 to 7 years, although ENSO itself also varies seasonally in intensity).

## 8 Modulation of the MJO by ENSO

Equipped with our implementation of the skeleton model suitable to incorporate time-dependent observation-based forcings, we now study changes in the statistics of selected MJO characteristics under the different phases of ENSO (El Niño, La Niña and neutral phase) in observations and in the model. To identify the ENSO phase during the period 1979-2021, we use the NOAA Oceanic Niño Index (ONI), based on SST anomalies in the Niño 3.4 region from $170°-120°$W and $5°$S-$5°$N. The index is computed by averaging the monthly values of Niño 3.4 SST anomalies with a running three-month window. An El Niño event
is declared when the index is greater than or equal to $0.5°$C for at least 5 consecutive values, *i.e.* 5 consecutive overlapping three-month seasons. A La Niña event is declared when the index is less than or equal to $-0.5°$C for at least 5 consecutive values. The Niño3.4 values are available on the website of NOAA National Centers for Environmental Information (https://origin. cpc.ncep.noaa.gov/products/analysis_monitoring/ensostuff/ONI_change.shtml). For observational data, it is straightforward to identify MJO events which occurred during each of these phases based on the date of the event. Precisely, the whole length
of the MJO event is considered. An event is said to have occurred during El Niño/La Niña/neutral ENSO, if more than half of the event has occurred during that specific phase. For the model, as explained in Section 5, simulations are run with forcing corresponding to the period 1979-2021, resulting in time-stamped outputs which are considered representative of this period. We can thus identify simulated MJO events occurring during El Niño, La Niña and neutral ENSO phases based on their dates. In addition, whereas in observations the number of MJO events is constrained to a single 43-years record, limiting the significance
of statistical studies, we realize 15 independent runs of the model, obtaining much larger samples and more robust statistical results. The total number of MJO events in observations and in the model runs during each phase of ENSO is reported in Table 1.

| ENSO phase | El Niño | La Niña | neutral | Total |
|---|---|---|---|---|
| Observed MJO events | 42 | 36 | 75 | 153 |
| Simulated MJO events | 205 | 228 | 547 | 980 |

**Table 1.** Number of MJO events in observations and in the model, during each phase of ENSO.





## 8.1 MJO activity across El Niño, La Niña and neutral ENSO

We first report the occurrence of MJO events across ENSO phases for observations and simulations. Table 2 shows the percent-
age of MJO active days during El Niño, La Niña and neutral ENSO for the period 1979-2021 (in observations and simulations).
Note that, for the simulations, the values correspond to a mean over the 15 independent runs, and the standard error of the mean
is indicated. In addition, the total number of days (irrespective of MJO activity) belonging to each of the three phases is indi-
cated in the last row. We observe that the proportion of MJO days approximately follows the proportion of the total number of
days belonging to each phase of ENSO both for observations and simulations, with the majority of events occurring during the
neutral phase of ENSO, and the minimum during El Niño. When comparing percentages in the first row with those in the last
row, we see that, in observations, El Niño and La Niña conditions seem to slightly favor the occurrence of MJO events (with
respect to what would be expected from the proportion of these ENSO phases). On the other hand, when comparing the second
row to the last, we see that, in the simulations, the opposite is observed: the neutral ENSO conditions seem to slightly favor the
occurrence of MJO events. This small discrepancy could be attributed to the design of the model, i.e. the omission of certain
mechanisms and interactions with other climate phenomena, though the differences are minor and may simply reflect limited
sample sizes.

| ENSO Phase | El Niño | La Niña | neutral |
|---|---|---|---|
| Observed active MJO days | 24.8% | 26.3% | 48.9% |
| Simulated active MJO Events | $21.7 \pm 1.5\%$ | $23.7 \pm 1.16\%$ | $54.6 \pm 1.9\%$ |
| Total number of days per phase | 3497 (22.8%) | 3924 (25.6%) | 7920 (51.6%) |

**Table 2.** Comparison of observed and simulated percentages of active MJO days across ENSO phases – percentages from the total of active
MJO days, where "active MJO days" are the days during MJO events as defined in Section 4. The last line represents the total number of
days (active and inactive MJO) in each phase of ENSO over the period 1979-2021.

## 8.2 ENSO modulation of MJO characteristics

### 8.2.1 Observations

Figure 9 shows the cumulative probability distribution of the MJO events' durations, total angle covered in (RMM1, RMM2)
phase space and maximum RMM value for observed events occurring during El Niño, La Niña and neutral periods. The
distributions show several differences. During El Niño, observed MJO events appear to have a shorter duration compared to
those occurring during La Niña and neutral ENSO [Fig. 9(a)]. In fact, the mean duration of MJO events during El Niño is 33
days, 40 days for La Niña and 36 days for the neutral phase of ENSO. Further, the maximum duration is 73 days for MJO
events during El Niño, 100 days during La Niña and 153 days during the neutral phase of ENSO. Similarly, during El Niño, the
total angle covered in RMM phase space by MJO event seems to be shorter [Fig. 9(b)], suggesting that events propagate over





shorter distances. In fact, the mean angle covered in RMM phase space by MJO events occurring during El Niño is $0.7 \cdot 2\pi$, while it is $0.8 \cdot 2\pi$ for la Niña and the neutral phase of ENSO. The maximum angle covered in RMM phase space is $2.0 \cdot 2\pi$ for events occurring during El Niño (meaning that they have propagated twice around the entire globe), $2.4 \cdot 2\pi$ for events occurring during La Niña and $3.4 \cdot 2\pi$ for events occurring during the neutral phase of ENSO. Note that previous studies have

reported that during El Niño periods, MJO events tend to propagate further eastward (Kessler, 2001; Tam & Lau, 2005; Pohl & Matthews, 2007). This does not contradict the present results since the methodologies and definition of MJO events vary between these studies. In addition, we consider here the total angle in RMM phase space and not the final location of MJO events. Finally, for the maxima of RMM amplitude [Fig. 9(c)], the differences in the cumulative distributions are slightly more intricate than for the other two characteristics. The mean is $2.5$ for events during El Niño, $2.6$ during the La Niña and $2.5$ during

the neutral phase. The maximum values are $4.6$ for El Niño, $4.0$ for La Niña, and $3.9$ for the neutral phase of ENSO. The mean and maximum values of all characteristics are summarized in Table 3.

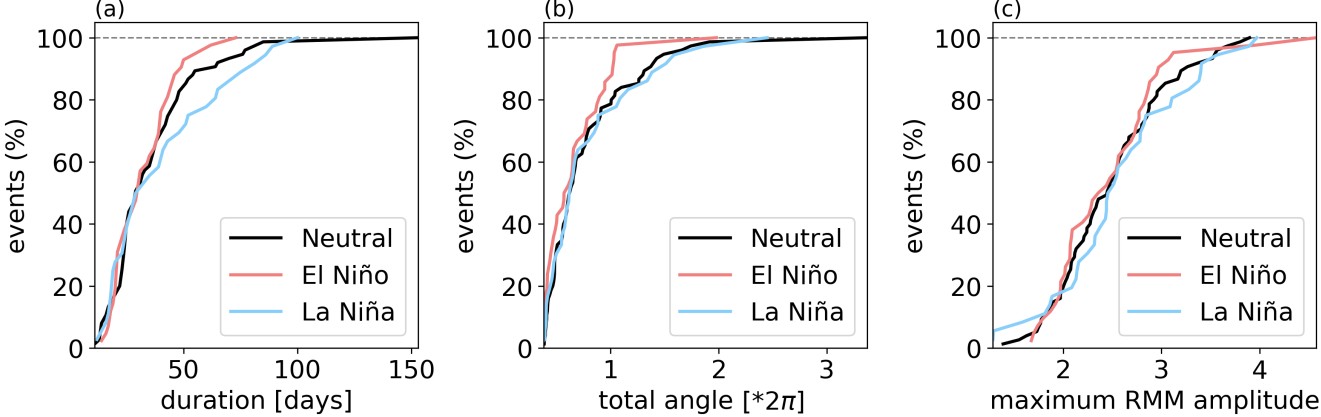

**Figure 9.** Cumulative probability distribution of the MJO events' durations, total angle covered in (RMM1, RMM2) phase space and maximum RMM value for *observed* events occurring during El Niño, La Niña and neutral periods between 1979 and 2021. The number of events in each sample is reported in Table 1.

### 8.2.2  Simulations

Finally, we look at the characteristics of MJO events in the model, during El Niño, La Niña and neutral conditions. The results are shown in Figure 10. For all three MJO characteristics, the cumulative distributions are very similar during El Niño, La Niña

and neutral conditions. We perform a Kolmogorov-Smirnov test to identify whether statistically significant differences exist between the El Niño and La Niña samples for each of the three characteristics. For the duration of MJO events, the test yields a $D$ value (representing the maximum distance between the two curves) of $0.0865$ and a $p$ value of $0.4$. For the total angles covered, the $D$ value is $0.1307$ and the $p$ value $0.05$. For the maxima of SMM values, the $D$ value is $0.0858$ and the $p$ value





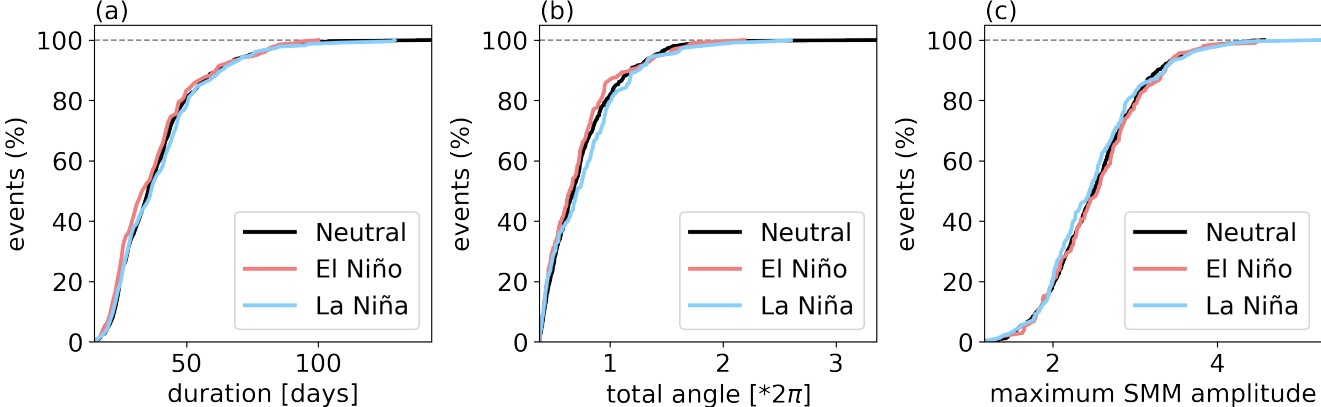

**Figure 10.** Cumulative probability distribution of the MJO events' durations, total angle covered in (SMM1, SMM2) phase space and maximum SMM value for *simulated* events occurring during El Niño, La Niña and neutral periods. The simulations were forced with data covering the period 1979-2021. The number of events in each sample is reported in Table 1.

0.4. Hence, none of the three tests provides sufficient evidence to reject the null hypothesis that the two samples come from the same distribution at significance level $0.05$.

We might nonetheless compare the mean and maxima of the three characteristics. To do so we compute these values in 15 independent simulation runs and determine the mean and standard error of the resulting values. They are summarized in Table 3. On average, we observe slightly smaller mean and maximum durations of MJO events during El Niño compared to other phases of ENSO. We also observe smaller mean and maximum values (on average) of the total angle covered in SMM phase space during El Niño. No specific tendency is observed for the maxima of SMM values. In all cases, the differences between these values remain small.

### 8.2.3 Discussion

Comparing observations and simulations, the main difference lies in the fact that, in the simulations, the cumulative distributions of MJO characteristics do not significantly differ under different ENSO conditions (unlike in observations). This could be due to the model's failure to accurately represent certain physical mechanisms or to a lack of sufficient observational data.

However, some similarities between observations and simulations are still seen when looking at the mean and maxima of MJO characteristics. In both observations and the model, the mean and maximum duration of MJO events are smaller for events occurring during El Niño and larger for events occurring during La Niña (although these differences are relatively small in the model). Similarly, the mean and maximum of total SMM/RMM angle are smaller during El Niño than during La Niña. On the other hand, no such general tendency is seen for the maximum of SMM/RMM amplitude.





| Parameter | Statistic | El Niño | La Niña | neutral |
|---|---|---|---|---|
| Duration (days) | Mean | 33 | 40 | 36 |
| | Max. | 73 | 100 | 153 |
| | Mean | $38.3 \pm 1.2$ | $40.7 \pm 1.9$ | $39.6 \pm 0.8$ |
| | Max. | $73.4 \pm 4.4$ | $80.9 \pm 6.2$ | $90.7 \pm 5.6$ |
| Angle in SMM Phase Space | Mean | $0.7 \cdot 2\pi$ | $0.8 \cdot 2\pi$ | $0.8 \cdot 2\pi$ |
| | Max. | $2.0 \cdot 2\pi$ | $2.4 \cdot 2\pi$ | $3.4 \cdot 2\pi$ |
| | Mean | $(0.73 \pm 0.02)2\pi$ | $(0.79 \pm 0.04)2\pi$ | $(0.75 \pm 0.01)2\pi$ |
| | Max. | $(1.44 \pm 0.10)2\pi$ | $(1.55 \pm 0.13)2\pi$ | $(1.75 \pm 0.15)2\pi$ |
| Maxima of SMM Amplitude | Mean | 2.5 | 2.6 | 2.5 |
| | Max. | 4.6 | 4.0 | 3.9 |
| | Mean | $2.57 \pm 0.04$ | $2.51 \pm 0.05$ | $2.53 \pm 0.02$ |
| | Max. | $3.71 \pm 0.11$ | $3.82 \pm 0.14$ | $3.96 \pm 0.09$ |

**Table 3.** Comparison of the mean and maximum values of selected MJO characteristics between observations (grey) and simulations (blue). For observations, the values are calculated as a mean over 15 independent simulation runs with the corresponding standard error.

## 9 Conclusions

We have implemented time-varying observation-based forcing profiles in the MJO skeleton model. As in previous works (Majda & Stechmann, 2009b, 2011; Thual et al., 2014), the model captures several important features of the MJO including its phase speed of around 5 m/s, its flat dispersion relation, its horizontal quadrupole vortex structure (not shown here) and the intermittent generation of MJO events. We saw that, when considering planetary scales, the climatology and variance of observed convective activity are in very good agreement. Using an RMM-like index in the model we were able to objectively identify MJO events and to study their characteristics. We showed that the model cannot reproduce well seasonal variations in the MJO occurrences. This could be due to the 3-month averaging of the forcing profiles which blurs the differences between seasons. The model fails to accurately replicate spatial variations in the MJO properties, such as its starting and ending phase. This limitation may be attributed to the stochasticity in the model, as pointed out in Stachnik et al. (2015). The stochasticity likely has the effect of dampening the geographic dependencies that should arise from the application of zonally varying forcing functions. Further investigation would be required to separate the effects of stochasticity, zonally varying forcing and nonlinearity on the spatial properties of simulated MJO events. On the other hand, similarities were found between observations and simulations for the statistics of MJO event durations, total angles covered in SMM/RMM phase space and maxima of SMM/RMM amplitude. To investigate how temporal variability at longer time scales affects the MJO, we evaluated differences in the statistics of these three MJO characteristics under the different phases of ENSO (El Niño, La Niña and neutral phase) in observations and in the model. We found that while observations might suggest some differences (for instance a tendency towards shorter-lived MJO events during El Niño), the model does not identify statistically significant differences in duration,




total angle covered in SMM phase space and in the maximum SMM value of simulated MJO events between the different
ENSO phases.

In conclusion, our results show that the model reproduces well the planetary-scale variability of convective activity and selected characteristics of MJO events, but it does not capture the impact of ENSO phases on these characteristics. Further investigation will be needed to determine whether the interannual variability of the model forcing functions impacts other MJO characteristics which have not been studied here, such as the propagation speed of individual events or their longitudinal extent.
In order to compute these characteristics as precisely as possible and to make objective comparisons between the model and observations, one will need to implement a method able to track the MJO (e.g. its convective center) both in the simulations and reanalysis.

Further, while the skeleton model allows to gain a better understanding of the fundamental physical mechanisms of the MJO, more complexity will likely be needed to fully reproduce the modulation of the MJO by ENSO. In particular, ENSO impacts
the extent of MJO penetration in the Pacific Ocean (Pohl & Matthews, 2007; Tam & Lau, 2005; Kessler, 2001). During El Niño years, MJO convective activities often extend eastward beyond the dateline into the Pacific, while during La Niña years, they tend to remain west of the dateline. Pohl & Matthews (2007) also reported that the duration of MJO events is longer during La Niña and shorter during El Niño when events occur from March to May and October to December. Hence, in order to be able to study the effects of ENSO on the MJO, one condition on any model should be that it captures the spatial variability of the
MJO as well as its seasonal variations.

*Code and data availability.* The datasets used for computing the model's forcing profiles (Section 3) are publicly accessible. The NCEP-NCAR reanalysis latent heat net flux data (Kalnay et al., 1996) can be freely downloaded from NOAA PSL, while the NCEP Global Precipitation Climatology Project (GPCP) data (Adler et al., 2017; Huffman et al., 2001) is available from the NOAA NCEI. The Python code used to compute these profiles is accessible on Ehstand (2025). The Julia implementations of the stochastic MJO skeleton model and
the code used for the postprocessing and analysis of its outputs are also available on Ehstand (2025), as well as the model output data and Julia notebooks to reproduce the figures in the present paper.

For observational indices: The RMM index data can be freely downloaded from the website of the Australian Bureau of Meteorology. The Niño3.4 index can be freely accessed on the website of NOAA (National Centers for Environmental Information).

*Author contributions.* N.E. performed the simulations, analyzed the data, created the figures, and wrote the first manuscript draft.
N.E., R.V.D., E.H.-G., C.L. directed the study. All authors contributed with ideas, interpretation of the results and manuscript revisions.

*Competing interests.* One of the authors is member of the editorial board of Nonlinear Processes in Geophysics



*Acknowledgements.* N.E. would like to thank Nan Chen and Tabea Gleiter for their help with understanding and implementing the stochastic skeleton model. She is also grateful to H. Reed Ogrosky and Samuel N. Stechmann for the helpful email exchanges. This project has received funding from the European Union's Horizon 2020 research and innovation programme under the Marie Skolodowska-Curie Grant

Agreement No 813844, from the Agencia Estatal de Investigación (MICIU/AEI/10.13039/501100011033) under the María de Maeztu project CEX2021-001164-M, and from the Agencia Estatal de Investigación (MICIU/AEI/10.13039/501100011033) and FEDER "Una manera de hacer Europa" under Project LAMARCA No. PID2021-123352OB-C32. R.V.D. acknowledges financial support by the German Federal Ministry for Education and Research (BMBF) via the JPI Climate/JPI Oceans Project ROADMAP (Grant No. 01LP2002B).



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
