# Peer review of "The stochastic skeleton model for the Madden-Julian Oscillation with time-dependent observation-based forcing"

_EGUsphere, 2025_

## Referee Comment (RC2)

**General Comments:**

The purpose of this paper is to introduce a time-dependent observation-based forcing term into the MJO skeleton model and analyze the characteristics of the MJO and its relationship with ENSO. However, throughout the paper, there is no clear evidence that the method used in this study improves the reproducibility of the MJO, and the contribution of the study is unclear.

**Major Comments:**

1. No evidence that time-dependent forcing has improved MJO simulation
   a. This study replaces traditional time-independent forcing with time-dependent forcing; however, there is no clear evidence that this modification has led to any improvement. In fact, there is even a possibility that the model's reproducibility has deteriorated.
   b. Specifically, the model fails to predict the timing of MJO events and does not capture seasonal variations.
   c. Moreover, applying a 3-month moving average to the forcing might have smoothed out important time variations, potentially leading to a failure in reproducing the MJO's temporal variability.
   d. These results contradict the stated research objectives, and the advantages of using time-dependent forcing remain unclear.

2. Agreement in MJO statistical characteristics but failure to predict MJO event timing
   a. While the study shows that the model reproduces statistical properties such as MJO duration and amplitude, it completely fails to predict the actual timing of MJO events. This raises serious concerns about the model's validity, as the agreement in statistical properties may not be meaningful.
   b. Since the model in this study fails to predict the timing of MJO events, the agreement in statistical characteristics could result from averaging effects rather than actual model skill. Presenting this as a validation of the model could be misleading.
   c. It remains unclear whether the agreement in statistical properties is a genuine consequence of incorporating time-dependent forcing or merely a result of parameter tuning to match observed statistics.

3. The model fails to capture ENSO-induced modulations of MJO events
   a. The model does not reproduce statistically significant differences in MJO characteristics across ENSO phases.
   b. This suggests that the model fails to properly capture ENSO's influence on MJO variability, contradicting the study's objectives.
   c. Compared to previous studies demonstrating clear ENSO-related MJO modulations, the results presented in this study appear insufficient in representing the expected ENSO influence.

4.  Lack of comparison with traditional models leaves improvements unverified
    a.  The manuscript does not clearly demonstrate what aspects have improved due to the introduction of time-dependent forcing.
    b.  Without a direct comparison to a time-independent forcing version of the model, it is impossible to assess the effectiveness of the proposed approach.

**Detailed Comments:**

1.  **Mathematical formatting:** Equations should be centered, with equation numbers aligned to the right, following standard academic formatting.
2.  **Line 55-56:** Specific results might be placed in later sections (results or discussion).
3.  **Line 58-63:** Is this about general SST variability or ENSO-driven SST variability? The transition between topics is unclear, making the focus ambiguous.
4.  **Line 89:** What modes were retained, and why? How does truncation impact model accuracy?
5.  **Line 138:** What advantages do these datasets have compared to other reanalysis datasets? Given the selected resolution, do you believe that the key characteristics of the MJO are adequately captured? How does this resolution affect the representation of small-scale variations in the MJO?
6.  **Line 178:** The 3-month averaging may remove key time variations, affecting MJO representation. How would results change with a shorter window (e.g., 1 month) or a different smoothing method?
7.  **Line 190:** Why not use RMM directly? How does excluding lower winds affect results? How well does SMM match RMM in identifying MJO? Since SMM fails to capture spatial variations, a direct comparison with RMM is necessary.
8.  **Line 211:** Please explain parameter selection. Were sensitivity tests conducted? If values are from past studies, cite references.
9.  **Line 250:** Please cite Figure 4c in the text. What causes variance overestimation? What was the basis for choosing the first 14 spatial modes? How important are high-wavenumber modes for MJO representation?

---

## Author Comment (AC1)

Dear Editor,

We acknowledge the effort the referees have taken to review our manuscript. Their constructive feedback has helped us improve the clarity and message of our work. Please find below our responses to the comments made by each referee. The corresponding changes in the manuscript appear in blue. We hope that our revisions address all concerns and that the manuscript is now suitable for publication.

**Response to Referee #1:**

We thank Referee #1 for his/her positive evaluation of our manuscript. We appreciate his/her recognition of the structure and clarity of our study, as well as the importance of incorporating time-dependent observation-based forcing into the MJO skeleton model, and that he/she finds our discussion of the model's strengths and limitations to be rigorous. Since Referee #1 recommends acceptance of the manuscript in its current form, no further modifications are needed in response to their comments.

**Response to Referee #2:**

We appreciate his/her detailed critical assessment of our work. Nevertheless, it seems that the objective of our work has not been properly captured. In this paper we have complemented the standard formulation of the stochastic skeleton model with time-dependent forcings computed from observation data. Our objective was to evaluate the model's ability, under this more realistic type of forcing, to reproduce key MJO characteristics and, in particular, its ability to reflect changes in these characteristics under the different phases of ENSO. To this end we have kept all parameter values of the original model at the values already established in the literature, without any additional tuning, and described in detail which aspects of the real phenomena are well described by the model with the new forcing and which are not. We have presented all the results obtained in a transparent manner, reporting both the ones for which the model performs well and the ones for which it shows limitations.

Below, we address each of the reviewer's concerns and clarify aspects of our study.

**1. No clear evidence that time-dependent forcing improves MJO simulation.**

a. This study replaces traditional time-independent forcing with time-dependent forcing; however, there is no clear evidence that this modification has led to any improvement. In fact, there is even a possibility that the model's reproducibility has deteriorated.

b. Specifically, the model fails to predict the timing of MJO events and does not capture seasonal variations.

c. Moreover, applying a 3-month moving average to the forcing might have smoothed out important time variations, potentially leading to a failure in reproducing the MJO's temporal variability.

d. These results contradict the stated research objectives, and the advantages of using time-dependent forcing remain unclear.

**Response:** Our study does not claim that time-dependent forcing improves MJO simulation. As mentioned above, the objective is *to evaluate* the model's ability, under a more realistic type of

forcing, to reproduce important MJO characteristics and to capture effects of seasonal and longer-term variability on these characteristics. We highlight that:

- The model successfully reproduces observed convective variance and climatology on planetary scales.

- We identify MJO events in the model using an RMM-like index (the SMM index) and show statistical agreement with observed event characteristics, including events' duration, total angle covered in RMM/SMM phase-space and maximum amplitude of the index. We did check that the reproducibility had not deteriorated over the time-independent model. We now include in Sect. 7.2 (lines 308-312) some numbers supporting that the inclusion of time-dependent forcing indeed improves (although in a small amount) the results with respect to the simulation with time-independent simulation.

- We highlight the limitations of the model in reproducing MJO's spatial variations and seasonality as well as the impact of ENSO on the chosen MJO's characteristics.

While the model does not fully capture seasonal variations in MJO occurrences, we discussed this limitation and provided proper explanations, including the effect of averaging over three-month periods. In fact, the 3-month average was applied to smooth out the monthly temporal variations of the forcing data. This helps capturing longer-term trends. However, this may neglect some important short-term variations. We have commented this limitation in the current version of the manuscript (in the Conclusion, lines 424-428) and suggest exploring shorter time averaging, or rather alternative methods, in future work.

**2. Agreement in MJO statistical characteristics but failure to predict individual event timing**

a. While the study shows that the model reproduces statistical properties such as MJO duration and amplitude, it completely fails to predict the actual timing of MJO events. This raises serious concerns about the model's validity, as the agreement in statistical properties may not be meaningful.

b. Since the model in this study fails to predict the timing of MJO events, the agreement in statistical characteristcs could result from averaging effects rather than actual model skill. Presenting this as a validation of the model could be misleading.

c. It remains unclear whether the agreement in statistical properties is a genuine consequence of incorporating time-dependent forcing or merely a result of parameter tuning to match observed statistics.

**Response:** We believe there may be a misunderstanding regarding the objective of our study. Neither the original model, nor our work, is designed to predict individual MJO events but to reproduce their statistical properties. Predicting the exact timing of MJO events is a challenge even for very complex ocean-atmosphere models well beyond the scope of the skeleton one. Our goal is instead to assess how well the stochastic skeleton model captures the statistical properties of MJO activity, which is a widely accepted approach in studies of low-order models as the skeleton model is. In addition, with respect to the comment by the referee on possible parameter tuning, we stress that we use the same parameters as in the previous implementation of the stochastic skeleton model with time-independent observation-based forcing (*Ogrosky & Stechmann, 2015*). Note the slight

difference of notation: Γ in Ogrosky and Stechman 2015 ↔ γ*Γ in our manuscript. No further parameter tuning has been performed, to show more clearly the effect of the new ingredient introduced here: the time-dependent forcing.

We have included precisions in the manuscript:

    a) In the introduction (lines 49-51)

    b) In Section 3 (lines 137-138)

    c) A citation of *Ogrosky & Stechmann (2015)* at the beginning of Section 5 (line 214)

**3. The model fails to capture ENSO-induced modulations of MJO events**

a. The model does not reproduce statistically significant differences in MJO characteristics across ENSO phases.

b. This suggests that the model fails to properly capture ENSO's influence on MJO variability, contradicting the study's objectives.

c. Compared to previous studies demonstrating clear ENSO-related MJO modulations, the results presented in this study appear insufficient in representing the expected ENSO influence.

**Response:** Certainly our model does not fully reproduce the observed modulation of the MJO by ENSO phases. Since this is a negative result, we could have chosen not to mention it. However, we believe that this negative result is useful for the community, as it highlights the limitations of the simplified structure of the stochastic skeleton model. We believe that properly reporting these limitations is not an argument against publication, but rather supporting it.  Previous studies *(Díaz et al., 2023; Suematsu & Miura, 2022; Wei & Ren, 2019)* have suggested that more complex interactions, such as those involving changes in mean state winds and ocean coupling, may be necessary to accurately capture ENSO's impact on the MJO. We confirm in the present study that this type of processes, beyond the simplicity of the skeleton model, is indeed needed to properly account for MJO modulation by ENSO.

We have added some sentences in the Conclusion (lines 440 – 443).

**4. Lack of comparison with traditional models leaves improvements unverified**

a. The manuscript does not clearly demonstrate what aspects have improved due to the introduction of time-dependent forcing.

b. Without a direct comparison to a time-independent forcing version of the model, it is impossible to assess the effectiveness of the proposed approach.

**Response:** Explicitly time-dependent features, such as MJO seasonality or the impacts of ENSO on the MJO, can only be studied with time-dependent forcing. Therefore, we could not compare such features with the time-independent version of the model.  Comparison of other MJO characteristics (event duration, angle traveled, SMM index amplitude) can be done with the time-independent-forcing implementations of previous studies, and brief comments on this have been added to our manuscript (Sect. 7.2, lines 308-312), as detailed in the response to point 1 by the referee. The

introduction of observation-based time-dependent forcing brings model results slightly closer to observations.

As mentioned previously, the introduction of time-dependent forcing in the model aims at evaluating the model's ability to capture effects of seasonal and longer-term variability on the MJO. The distributions of MJO events durations, SMM angles and SMM amplitudes are already well represented in the model with time-independent forcings. Therefore, although our version of the model's forcing slightly improves the statistical reproduction of such characteristics, this was not the primary aim. This is why we chose to compare the model's outputs against observations (to evaluate its performance in reproducing real conditions), and not performing an exhaustive comparison with a previous model's version (which would have been needed to assess improvement over that version).

**5. Minor comments.**

1. Mathematical formatting: Equations should be centered, with equation numbers aligned to the right, following standard academic formatting.

The Copernicus official template sets the equations with a small left indent rather than centering them (line '`\LoadClass[fleqn]{article}`'in the '`copernicus.cls`' file). We thus have to keep the current formatting.

2. Line 55-56: Specific results might be placed in later sections (results or discussion)

We removed the following sentence from the introduction and kept the description of these specific results' description in Sections 7:

*We show that several properties of the simulated events (in particular, their lifetime, extent and amplitude) agree well with the ones of observed MJO events. On the other hand, we find that the seasonality of MJO events and the spatial variations in the MJO properties are not well reproduced by the model.*

3. Line 58-63: Is this about general SST variability or ENSO-driven SST variability? The transition between topics is unclear, making the focus ambiguous.

We agree that the formulation was ambiguous. We clarified it by rewriting the sentence as (lines 58-61):

*Further, several studies suggested that the sea surface temperature (SST) variability at interannual and longer time scales influences the MJO's characteristics. In particular the variability associated with the El Niño--Southern Oscillation (ENSO) modulates the extent of MJO's eastward propagation (Kessler, 2001; Tam & Lau, 2005; Pohl & Matthews, 2007), its lifetime (Pohl & Matthews, 2007) and its speed (Wei & Ren, 2019; Díaz et al., 2023).*

4. Line 89: What modes were retained, and why? How does truncation impact model accuracy?

In our study, we follow the standard truncation procedure described in the original papers on the model *(Majda & Stechmann, 2009b, 2011; Thual et al., 2014).* One of the aims of the skeleton theory is to propose a minimal set of equations able to capture key features of the MJO. The truncation is also key to facilitate the model's implementation.

For the vertical truncation only the first baroclinic mode is retained. In the meridional direction, the variables and forcing functions are expanded in a basis of parabolic cylinder functions. This expansion facilitates a change of variable in the dry dynamics which allows to introduce new variables representing equatorial waves (Kelvin waves, mixed Rossby-gravity waves and Rossby waves). Only the first mode of equatorial Kelvin wave structure (K) and equatorial Rossby wave structure (R) are kept (which might be written in terms of the first three parabolic cylinder functions). We refer to *Majda et al. 2019* [Section 2.3.3] for full details. This citation was also added to the manuscript.

5. Line 138: What advantages do these datasets have compared to other reanalysis datasets? Given the selected resolution, do you believe that the key characteristics of the MJO are adequately captured? How does this resolution affect the representation of small-scale variations in the MJO?

We used these reanalysis datasets following *Ogrosky & Stechmann (2015)*. This allowed us to compare our initial estimation of the time-independent forcing terms to theirs. We believe that the resolution of the datasets are adequate for the MJO skeleton model since its resolution is much lower than that of the datasets: the model has 64 points around the equator corresponding to a 5.6° resolution. The model is in fact designed to capture fundamental features of the MJO on planetary-scales only.

The spatial and temporal resolution of the model were clarified in the beginning of Section 5 (lines 215-217).

6. Line 178: The 3-month averaging may remove key time variations, affecting MJO representation. How would results change with a shorter window (e.g., 1 month) or a different smoothing method?

When using a 1-month smoothing, the number of MJO events decreased radically in the simulations. As mentioned in the manuscript, we believe that this might be due to sharper variations in the forcings disrupting the model's statistical equilibrium. A more thorough investigation would be required to fully explain this behavior, but it is beyond the scope of our study.

7. Line 190: Why not use RMM directly? How does excluding lower winds affect results? How well does SMM match RMM in identifying MJO? Since SMM fails to capture spatial variations, a direct comparison with RMM is necessary.

The RMM index uses three variables (upper and lower tropospheric wind as well as OLR). These variables are not directly available in the model. Thus, RMM cannot be computed in the model. Nonetheless, as explained in *Stachnik et al. (2015),* a similar index (SMM index) can be computed by using the zonal wind as a substitute for the lower tropospheric wind and the convective activity variable (bar{H}a) as a proxy for OLR. Upper winds are excluded but, as stated in the original paper: "the choice of a bivariate index does not preclude a meaningful comparison of the skeleton model to observations". The authors cite *Straub (2013),* who found a high correlation between the full RMM index and RMM-like indices calculated without upper tropospheric winds.

The SMM index was effectively designed to facilitate comparison between the skeleton model and observational data (*Stachnik et al. 2015*).

*Straub, K. H., 2013: MJO Initiation in the Real-Time Multivariate MJO Index. J. Climate, 26, 1130–1151, https://doi.org/10.1175/JCLI-D-12-00074.1.*

8. Line 211: Please explain parameter selection. Were sensitivity tests conducted? If values are from past studies, cite references.

As mentioned in the above responses, parameters were taken from *Ogrosky & Stechmann (2015)*. The references is now cited in Section 5 (line 214).  Some sensitivity tests were performed in *Majda & Stechmann (2009)* for the time-independent linear version of the model.

9. Line 250: Please cite Figure 4c in the text. What causes variance overestimation? What was the basis for choosing the first 14 spatial modes? How important are high-wavenumber modes for MJO representation?

The model is initially constructed to include only large-scale MJO's freatures (the MJO's envelope has a scale of the order of 10000km). However, the stochastic version of *Thual et al. (2014)* that we implement here introduces fluctuations at all spatial scales, down to the grid size of 625 km, which the model is not able to completely filter out. This is also seen in the unfiltered panels of Fig. 2. The resulting excess of variance is evident in Fig. 4c, but it does not affect the spatial profile of the temporal average of the convective activity (Fig. 4a). We empirically find out that filtering out all the spatial modes except the 14 ones with longest wavelength correctly reproduces variance observational data. We do not have a full theoretical justification for the suitability of this particular number of modes, and this is now explicitly stated when commenting on Fig. 4c.

In addition, as mentioned above a clarification of the model's spatial scales and a note on the stochastic parametrization have been added at the beginning of Section 5 (lines 215-218).

---

## Referee Report (RR1)

**General Comments:**

This manuscript represents an important advancement by incorporating observation-based, time-dependent forcing into the stochastic skeleton model of the Madden-Julian Oscillation (MJO). It demonstrates effectively that the model reproduces key statistical characteristics of the MJO, clearly outlining the model's capabilities and limitations, especially regarding seasonal variations and ENSO modulation. However, to strengthen the conclusions and scientific rigor of the study, additional discussions and analyses are necessary, particularly regarding the following points:

**Major Comments:**

1. Validity and Impact of the 3-month Moving Average

   The use of a 3-month moving average for smoothing the forcing profiles significantly influences the results. The authors acknowledge that this smoothing may be responsible for the inadequate representation of seasonality. Thus, further in-depth discussions are required:

   - Rationale for choosing a 3-month window: Provide clear theoretical or empirical justifications for this specific smoothing window. Stating merely that it captures "long-term trends" is insufficient. Discuss explicitly how results might differ if shorter or longer smoothing windows (e.g., 1-month or 6-month averages) were employed.

   - Necessity of sensitivity experiments: Ideally, sensitivity experiments with different smoothing windows (e.g., no smoothing, 1-month average) should be conducted and presented, at least as supplementary materials. Such analysis would quantitatively reveal how significantly this smoothing affects seasonal and interannual signals.

   Without this analysis, the conclusion that observation-based forcing enhances realism remains insufficiently supported.

2. Diagnosing the Lack of ENSO Modulation

   A key finding is the model's inability to reproduce the observed modulation of MJO characteristics under different ENSO phases. The authors plausibly attribute this to the model's structural simplicity, such as the lack of ocean coupling and mean-state wind interactions. However, to strengthen this conclusion, it is essential to first diagnose the source of this discrepancy. Another possibility is that the 1D forcing profiles themselves do not adequately represent the ENSO signal.

I strongly recommend that the authors analyze the forcing functions (Sq and Sθ) to determine if they contain a statistically significant ENSO signal. This can be done by comparing the forcing profiles during El Niño, La Niña, and neutral periods.

- If the forcing does contain a clear ENSO signal, yet the model fails to respond, this would strongly support the authors' conclusion about the model's structural deficiencies.
- Conversely, if the forcing signal is weak or absent, it would suggest that the limitation lies within the forcing methodology itself (i.e., the projection of 2D observational data onto a 1D profile).
- This analysis would allow for a much clearer separation of causes and significantly enhance the scientific impact of the findings.

3. Statistical Evaluation and p-value Interpretation (Lines 314–320)

The current statistical interpretation confusing. Typically, a significance level of 5% (p=0.05) is standard, yet the manuscript mentions a level of 0.04 without clear justification.

The choice of a 0.04 significance level seems arbitrary and post-hoc. The authors should evaluate this result against the standard $p < 0.05$ threshold or provide a clear justification for their non-standard choice.

---

## Author Response (AR2)

We thank both reviewers for their time and effort in revising our manuscript and for their constructive feedback.

*Referee #1* recommends acceptance of the manuscript in its current form.

We have revised the manuscript and addressed all comments raised by *Referee #2*. Our responses are detailed below. All changes have been highlighted in blue-green in the revised version of the paper.

**1. Validity and Impact of the 3-month Moving Average**

When shorter averaging windows are used (e.g. 1 month), the number of MJO events produced by the model sharply drops. This is illustrated in the figure below, showing the mean number of MJO events per year as a function of the smoothing/averaging window. The dashed line corresponds to the mean number of events produced by the model with time-independent forcing (infinite averaging window). Note that while the manuscript reports statistics based on 15 independent simulation runs (to obtain larger samples and more robust statistics), only a single model run was used to produce the figure below.

The model thus seems to require a certain degree of smoothness or persistence (in duration and amplitude) in the forcing to generate MJO events. The chosen 3-month-window averaging seems to be appropriate to represent a range of window sizes suitable to generate MJO events, while still maintaining a time-dependence of the filtered forcing below seasonal or annual scales, in accordance with the objectives of this paper. We feel that performing a more exhaustive sensitivity analysis with more window sizes and increased statistics would be beyond the scope of this study. Specially because we do not expect the model's minimalistic structure to capture all aspects real world complexity.

We clarified the reason for the choice of a 3-month averaging window at the end of Section 3 and in the conclusion.

[Figure]

**2. Diagnosing the Lack of ENSO Modulation**

Following the reviewer's recommendation, we compared the forcing functions during El Niño, La Niña and the neutral periods. The figure below shows the mean Sq and Stheta profiles during each phase of ENSO (lines). The shaded areas correspond to 1 standard deviation around the means. The largest differences are found in the eastern equatorial Pacific (around the NIÑO 3 region, 210°E-270°E).
We used the Kolmogorov-Smirnov test to assess whether the distributions of Sq and Stheta values during El Niño and La Niña differed significantly at each spatial point. We found differences at nearly all points (at significance level 0.05). Locations with significant differences are indicated by stars at the bottom of the figure.
These observations support the argument that the model's lack of response to the ENSO signal reflects structural deficiencies within its design.

We added these figures in the paper, and commented them in the beginning of Section 8 and in section 8.2.3.

[Figure]

[Figure]

**3. Statistical Evaluation and p-value Interpretation**
We agree with the reviewer's comment and modified the relevant paragraph in Section 7.2, evaluating the results against the standard $p < 0.05$.
We also soften slightly the conclusion of Section 7, regarding the agreement in statistical features.

---

## Author Response (AR3)

**LIST OF CHANGES:**

The only difference in the files of this upload (21 Jul 2025) with respect to the editorially accepted previous version is the requested inclusion of the sentence *"At least one of the (co-)authors is a member of the editorial board of Nonlinear Processes in Geophysics"* in the *"Competing interests"* section.